# Sharing Vitamin B_12_ between Bacteria and Microalgae Does Not Systematically Occur: Case Study of the Haptophyte *Tisochrysis lutea*

**DOI:** 10.3390/microorganisms10071337

**Published:** 2022-07-01

**Authors:** Charlotte Nef, Simon Dittami, Raymond Kaas, Enora Briand, Cyril Noël, Francis Mairet, Matthieu Garnier

**Affiliations:** 1Physiologie et Biotechnologie des Algues, IFREMER, Rue de l’Ile d’Yeu, F-44311 Nantes, France; raymond.kaas@free.fr; 2Station Biologique de Roscoff, Integrative Biology of Marine Models Laboratory, CNRS, Sorbonne University, F-29680 Roscoff, France; simon.dittami@sb-roscoff.fr; 3GENALG, PHYTOX, IFREMER, F-44000 Nantes, France; enora.briand@ifremer.fr (E.B.); matthieu.garnier@ifremer.fr (M.G.); 4SEBIMER, IRSI, IFREMER, F-29280 Brest, France; cyril.noel@ifremer.fr; 5PHYSALG, PHYTOX, IFREMER, F-44000 Nantes, France; francis.mairet@ifremer.fr

**Keywords:** microbial interactions, vitamin B_12_, cobalamin, haptophytes, *Tisochrysis lutea*, phytoplankton

## Abstract

Haptophyte microalgae are key contributors to microbial communities in many environments. It has been proposed recently that members of this group would be virtually all dependent on vitamin B_12_ (cobalamin), an enzymatic cofactor produced only by some bacteria and archaea. Here, we examined the processes of vitamin B_12_ acquisition by haptophytes. We tested whether co-cultivating the model species *Tisochrysis lutea* with B_12_-producing bacteria in vitamin-deprived conditions would allow the microalga to overcome B_12_ deprivation. While *T. lutea* can grow by scavenging vitamin B_12_ from bacterial extracts, co-culture experiments showed that the algae did not receive B_12_ from its associated bacteria, despite bacteria/algae ratios supposedly being sufficient to allow enough vitamin production. Since other studies reported mutualistic algae–bacteria interactions for cobalamin, these results question the specificity of such associations. Finally, cultivating *T. lutea* with a complex bacterial consortium in the absence of the vitamin partially rescued its growth, highlighting the importance of microbial interactions and diversity. This work suggests that direct sharing of vitamin B_12_ is specific to each species pair and that algae in complex natural communities can acquire it indirectly by other mechanisms (e.g., after bacterial lysis).

## 1. Introduction

Group B vitamins are essential cofactors required for the activity of enzymes in central metabolism [1,2]. Most microalgae in culture require at least one or a combination of vitamin B_1_ (thiamine), vitamin B_7_ (biotin), and vitamin B_12_ (cobalamin) [3,4,5], meaning that they must acquire these coenzymes from exogenous environmental sources. Cobalamin amendment experiments in several ecosystems demonstrated that the availability of this vitamin significantly limited phytoplankton growth in the Atlantic Ocean [6], in coastal environments [7], in the Gulf of Alaska [8], and in the Southern Ocean [9]. Vitamin B_12_ is the most complex vitamin known to date, and its synthesis is restricted to select species of bacteria and archaea [2,10,11,12]. In marine environments, several prokaryotic groups have gathered species able to produce vitamin B_12_, such as the bacterial genus *Halomonas* [13], for which representatives are able to sustain microalgae with the vitamin [4]. A study moreover reported that all species of the *Rhodobacteraceae* clade encoded vitamin B_12_ biosynthesis genes [14]. In Archaea, all sequenced species from the Thaumarchaeota lineage have the genetic equipment to produce cobalamin de novo [12].

Interkingdom interactions between microalgae and bacteria for B_12_ have been reported in the literature for instance for chlorophytes [15], where a facultative interaction between the freshwater alga *Lobomonas rostrata* and the bacterium *Mesorhizobium loti* was observed. In this system, algae delivered organic carbon, probably amino acids [16], to bacteria in return for cobalamin. In addition, mutualistic complementation of B vitamins between the marine chlorophyte *Ostreococcus tauri* (B_12_ and B_1_ auxotroph) associated with *Dinoroseobacter shibae* (*Roseobacteraceae*; B_3_, B_7_, and B_9_ auxotroph) has been described [17]. Regarding dinoflagellates, several authors [18,19,20] showed that a majority of them were cobalamin-dependent due to the presence of a B_12_-dependent methionine synthase gene *metH*, and the lack of a complete *metE* isoform. Some of them may rely on metabolic exchanges with associated bacteria to obtain the vitamin [21].

We previously proposed that haptophytes are exclusively cobalamin-dependent microalgae [22], meaning that this taxon must rely on an external source for B_12_ acquisition. Members of the Haptophyta are globally distributed [23,24] and significantly contribute to phytoplankton communities [25,26] as well as global carbon [27,28] and sulfur [29,30] cycles. The role of vitamin B_12_ in the physiology of this ecologically important group has been recently investigated [19,22]. However, how these microalgae acquire this cofactor remains poorly understood. In this context, we aimed to test whether haptophytes would obtain bacterial cobalamin in a reciprocal relationship with direct interactions or by developing on cellular extract. First, we investigated the ability of B_12_ production in twelve pre-selected strains belonging to *Halomonas* and *Phaeobacter* genera. Two strains were then selected for co-culture experiments with the model haptophyte species *Tisochrysis lutea*, and cultures with bacterial extracts were carried out. A final co-culture experiment involving a complex natural bacterial consortium was undertaken.

## 2. Materials and Methods

### 2.1. Bacterial Screening

A screening of vitamin B_12_-producing bacteria in our collections was conducted by focusing on the marine genus *Halomonas* and on species from the *Rhodobacteraceae* family, which are known to synthesize the vitamin in amounts sufficient to sustain microalgae [4,13,14]. The screened bacterial strains were isolated from marine microalgal cultures of our own laboratory collection (for *Halomonas* spp. strains 09-003, 09-027, 09-213, 09-729, and *Phaeobacter* sp. strain 09-029) [31] or from deep-sea ecosystems (*Halomonas* spp. strains 1211, 1212, 1216, 1229, 1233, and 1236) (Laboratory EM3B, Ifremer, Brest, France). *Halomonas* spp. strain 19-001 was a gift from Prof. Alison Smith, University of Cambridge, UK. A total of twelve strains were considered (Table 1).

Bacteria were grown in three biological replicates in liquid Marine Broth medium (BD Difco™, Becton Dickinson Company, Franklin Lakes, NJ, USA) for 24 h at 28 °C with shaking (180 rpm). Cellular concentration was quantified by plating on Marine Agar in serial dilutions. Cells and supernatants were separated by centrifugation (10,000× *g*, 10 min 10 °C). Particular vitamin B_12_ extraction was conducted by referring to the protocol described in [22], by briefly boiling cells in PBS buffer for 15 min and collecting the supernatant after centrifugation. An effort was made to standardize the number of cells measured for their B_12_ quota. Vitamin B_12_ in the dissolved fraction was pre-concentrated by centrifugation using a 1000 Da column (Microsep Advance, Pall Corporation, New York, NY, USA). To avoid any bias caused by the presence of cobalamin in the culture medium, a Marine Broth sample was also considered for the B_12_ assay. Final B_12_ quantification in both particulate and dissolved fractions was realized with an ELISA test kit (Immunolab GmbH, Kassel, Germany) by referring to the provided protocol.

### 2.2. Bacterial Genome Sequencing

Four bacterial strains (*Halomonas* sp. 09-003, 09-027, and 19-001 as well as *Phaeobacter* sp. 09-029) were selected for genome sequencing. Strains 19-001 and 09-029 were chosen as they showed the most elevated B_12_ quota (see Figure 1). Strains 09-003 and 09-027 were also sequenced to assess the diversity of B_12_-related enzymes of *Halomonas* bacteria isolated from microalgal cultures. DNA extraction was performed by adding 530 µL of Tris EDTA HCl pH 7.5, 60 µL of SDS 10% and 3 µL of proteinase K (20 mg/mL) to a cellular pellet and incubating 90 min at 55 °C with shaking. Nucleic acids were purified by adding 100 µL of NaCl solution (5 M) and one volume phenol/chloroform isoamyl alcohol (25:24:1), centrifuged twice to harvest the upper phase, followed by a last purification step with chloroform. DNA was precipitated by adding 0.8 volume isopropanol and placing samples in ice for one hour. The DNA pellets were harvested by centrifugation then washed with ethanol 70% and finally suspended in 100 µL of Tris EDTA HCl (10 mM, pH 7.5).

The sequencing step was realized at the Roscoff Biological Station (France). Paired-end DNA libraries with an average insert size of 500 bp were prepared using the Nextera XT kit following the manufacturer’s protocol. Libraries were then sequenced together with other microbial genomes using the Illumina MiSeq technology (V3, paired end, 2 × 300 bp reads) at the GENOMER platform (Station Biologique de Roscoff, Nantes, France), multiplexing 20 bacterial genomes per run. Raw reads were first examined using FastQC (https://www.bioinformatics.babraham.ac.uk/projects/fastqc/ (accessed on 1 August 2019)). Low-quality sequences were trimmed or removed using Trimmomatic v.0.38 [32], using a sliding window with a quality score of 15 and a minimal read length of 36 bp as filters. Trimmed read pairs were used for genome assembly with SPAdes v.3.12 [33] using default parameters. Genomic sequences encoding ribosomal genes were identified using Barrnap v.0.8 (https://github.com/tseemann/barrnap (accessed on 1 August 2019)) and 16S rDNA sequences were used to search for complete reference genomes in the GenBank. These reference genomes (the GenBank identifiers were LN813019 for 19-001; CP024811.1, CP007757.1, CP034367.1, and CP020562.1 for both 09-003 and 09-027; CP016364.1 CP010588.1, CP010855.1, and NC_023135.1 for 09-029) were used for scaffolding with Medusa version 1.6 [34]. Finally, gaps in the scaffolds were filled wherever possible using GapCloser 1.12 [35] and the resulting draft genomes were annotated using the MicroScope platform [36]. The resulting genomes are available under the accession identifiers: “*Halomonas alkaliphila* PBA_09_003”; “*Halomonas alkaliphila* PBA_09_027”; “*Halomonas* sp. PBA_19_001” and “*Phaeobacter porticola* PBA_09_029”. Genes involved in B_12_ transport, metabolism, or biosynthesis were all searched for in this platform.

### 2.3. Phylogenetic Analysis of the Bacteria

The phylogenetic analysis was conducted using the Phylogeny web servers (http://www.phylogeny.fr/ (accessed on 5 May 2022)). Sequence alignment for the 16S rDNA gene was realized with MUSCLE v3.8.31 [37] in full mode for 50 bacterial sequences of which 46 were reference strain sequences of Alpha- and Gammaproteobacteria from the NCBI database. Sequences ranged between ~1400 and 2000 nucleotides. The curation step was conducted using the Gblocks v0.91b [38] tool and allowing gap positions within the final blocks. A total of 65% (1290) of initial positions were conserved. The final phylogenetic tree was constructed by maximum likelihood using PhyML v3.1 [39] (100 bootstraps) with default parameters, corresponding to the GTR substitution model.

### 2.4. T. lutea–Bacteria Co-Cultures and Analyses

For each condition, culture experiments were carried out in three biological replicates in 1 L autoclaved glass bottles homogenized by bubbling of 0.2 µm filtered air and maintained at 27 ± 1 °C with continuous light (150 µmol photons m^−2^ s^−1^). Sampling and cell enumeration of microalgal cultures were performed on a daily basis with a Multisizer Coulter Counter (Beckman Coulter, Brea, CA, USA). Maximal biomass increase (ΔC_max_) was estimated as ΔCmax = X_f_ − X_i_; where X_f_ and X_i_ are algal concentrations at the stationary phase and at the beginning of the culture, respectively. Vitamin B_12_, when added to the culture medium, was added in the form of cyanocobalamin (Sigma-Aldrich, Dorset, UK).

First, axenic pre-cultures of *Tisochrysis lutea* (CCAP 927/14), a tropical haptophyte isolated from the Pacific Ocean [40] (see [22,41] for the purification protocol), were cultivated in modified Conway medium [42] with 40 ng/L B_12_, a concentration known to provide relatively high biomass without being in excess in *T. lutea* cultures [19]. These pre-cultures were then transferred after three weeks of growth in fresh medium with 40 ng/L B_12_. This step was repeated two times. Final axenic pre-cultures were considered to inoculate two parallel experiments to evaluate the ability of *T. lutea* to use B_12_ from bacteria: 

On the one hand, we started by cultivating control axenic *T. lutea* with or without B_12_ addition. Then, starting from a fraction of the B_12_-deprived axenic controls in stationary phase collected on day 35, we cultivated *T. lutea* with bacterial extracts from the B_12_-producing *Phaeobacter* 09-029 and non B_12_-producing *Halomonas* 09-729 as control. Bacterial extracts were obtained with twenty-four hours aged bacterial cultures harvested by centrifugation and lysed by boiling for 15 min in 1 mL autoclaved ultrapure water, as was done previously [4,22]. The equivalent of 100 bacteria/alga extract for 09-029, corresponding to 70 mg/L carbon in final *T. lutea* cultures, was added to each of the three replicates. The quantity of 09-729 bacterial cells to be lysed was normalized on the same carbon basis than that of 09-029 (i.e., ~80 bacteria/alga). Vitamin B_12_ concentrations in the bacterial extracts were measured as above.

On the other hand, we prepared co-cultures with live bacteria. Bacteria and algae were first co-acclimated to initiate potential biological interactions. First, 1 L of axenic *Tisochrysis lutea* maintained in modified Conway medium [42] with 40 ng/L B_12_ was inoculated with 10 µL of a bacterial suspension. Both pre-cultures and cultures were done in similar conditions as controls. For the co-culture experiments themselves, the pre-cultures were transferred after one week in stationary phase either to B_12_-deprived (no cobalamin added) or B_12_-enriched (40 ng/L) medium at a concentration of 0.1 × 10^6^ *T. lutea* cells/mL, with bacterial ratios between ~3–5 bacteria/alga (see Section 3). Bacterial cells were enumerated once a week by flux cytometry (BD Accuri C6, BD Biosciences, San Jose, CA, USA) with 1% Sybr™Green (Lonza, Basel, Switzerland) staining to stain DNA [43]. Axenicity of control cultures was verified by visual microscopic inspection and standard plating analyses.

Next, to test whether a complex microbial community would be able to rescue the growth of *T. lutea* in B_12_-deprived culture conditions, we set up an experiment starting with a xenic mother culture of *T. lutea* (CCAP 927/14) that was maintained in the laboratory collection without previous purification. Axenic control cultures were set up as above. Both xenic and axenic pre-cultures were maintained and transferred several times in B_12_-enriched medium (40 ng/L B_12_) as was done for the bispecific co-culture experiment. Final cultures were inoculated with ~0.5 × 10^6^ algal cells/mL.

### 2.5. Identification of the Microbial Consortium

Genomic DNA from the algal-bacterial consortium was extracted following the DNA Nucleospin Plant II extraction kit protocol (Macherey-Nagel, Hoerdt, France). Bacteria were examined with primers targeting the V3-V4 hypervariable region of the 16S SSU rDNA gene using universal primers assembled with the Illumina adapters: PCR1F_460 5′CTT-TCC-CTA-CAC-GAC-GCT-CTT-CCG-ATC-TAC-GGR-AGG-CAG-CAG’3 and PCR1R_460 5′GGA-GTT-CAG-ACG-TGT-GCT-CTT-CCG-ATC-TTA-CCA-GGG-TAT-CTA-ATC-CT’3 [44,45,46]. Each PCR reaction, performed in triplicate, contained ~10 ng of extracted DNA, 0.2 µM of each primer, 1× Taq polymerase (Phusion High-Fidelity PCR Master Mix with GC Buffer) and DNA/RNAse free water for a total volume of 25 µL. The PCR cycle conditions included an initial denaturation of 5 min at 95 °C, 30 cycles of denaturation for 10 s at 95 °C, annealing for 30 s at 65 °C, and elongation for 20 s at 72 °C, one post-elongation step for 5 min at 72 °C, and then 4 °C. PCR products quality and integrity were verified by 1% agarose gel electrophoresis, and triplicates were then pooled. Secondary PCR amplification for the addition of the Illumina compatible sequencing adapters and unique per-sample indexes was conducted at GeT-PlaGe France Genomics sequencing platform (Toulouse, France). Barcoded amplicons were quantified, quality-checked, normalized, pooled, and sequenced within one sequencing run using the 2 × 250 paired-end method on an Illumina MiSeq instrument with a MiSeq Reagent Kit V3 chemistry (Illumina), according to the manufacturer’s recommendations. Raw data were analyzed using the SAMBA v3.0.1 workflow (https://github.com/ifremer-bioinformatics/samba (accessed on 5 May 2022)); [47]), a Standardized and Automatized MetaBarcoding Analysis workflow using DADA2 [48] and QIIME2 [49] with default parameters unless otherwise indicated. This workflow developed by the SeBiMER (Ifremer Bioinformatics Core Facility) is an open-source modular workflow to process eDNA metabarcoding data. SAMBA was developed using the NextFlow workflow manager [50] and built around three main parts: data integrity checking, bioinformatics processes, and statistical analyses. Firstly, a SAMBA checking process allows one to verify the raw data integrity. Sequencing primers were then trimmed from reads and reads where primers were not found were removed. Then, DADA2 was used to filter bad quality reads, correct sequencing errors, overlap paired reads, infer Amplicon Sequence Variants (ASVs), and remove chimeras. Due to the known diversity overestimation generated by DADA2, an additional step of ASV clustering (Operational Taxonomic Unit (OTU) calling) was performed using the dbOTU3 algorithm [51]. A sample decontamination step was conducted using microDecon [52] in order to remove ASVs present in both the control and biological samples. Taxonomic classification was achieved using the SILVA 138 [53,54] reference database. Following identification, reads detected in at least two of the three replicates were kept. Functional predictions were inferred using PICRUST2 [55] to draw an approximate picture of the metabolic pathways encoded within the microbial community.

### 2.6. Statistical Analyses

Two-way ANOVA tests were performed to compare the maximal biomass increase (ΔC_max_) of the co-culture experiment with either live bacteria or bacterial extracts. A two-tailed Student’s *t* test was conducted to compare the same parameter for the co-culture experiment with the microbial consortium. All statistics and figures were computed on RStudio v4.1.1 (RStudio Team, PBC, Boston, MA, USA).

## 3. Results

### 3.1. Selection of Vitamin B_12_-Producing Bacteria

Several studies reported vitamin B_12_ synthesis in marine bacteria from the *Halomonas* genus and the *Rhodobacteraceae* family [4,13,14]. We therefore investigated vitamin B_12_ production of different bacterial strains originating from different laboratory collections, either isolated from microalgal cultures or from hydrothermal vents and previously identified as *Halomonas* sp., together with one bacterium isolated from microalgal culture and identified as *Phaeobacter* sp. The twelve selected bacteria were screened in both intracellular and extracellular compartments to assess whether they contained and/or excreted B_12_ in simple monocultures without microalgae. As bacteria were grown in Marine Broth medium containing vitamin yeast extract and therefore B_12_, we measured the medium B_12_ (~157 ng B_12_/L) and subtracted it from the total concentration (sum of intracellular and extracellular vitamin). All bacterial strains investigated but two (strains 1236 and 09-729), produced intracellular vitamin B_12_, with concentrations ranging from 2 to 16 × 10^−20^ g B_12_/cell (Figure 1). Extracellular vitamin was not detected for any strain.

### 3.2. Genomic Analyses of the B_12_-Producing Bacteria

Based on the screening results, four B_12_-producing bacteria (*Halomonas* sp. 09-003, 09-027, 19-001; and *Phaeobacter* sp. 09-029) were selected for further genome sequencing to identify genes and metabolic pathways related to B_12_ production and dependency. First, the phylogenetic analysis of 16S rDNA sequences from genomes allowed us to determine the taxonomic position of each strain (Appendix A). Strain 09-029 was found to be closely related to *Sulfitobacter* spp. and *Phaeobacter* sp., and was identified as *Phaeobacter porticola* after deposition of the full sequenced genome in the MicroScope platform (https://mage.genoscope.cns.fr/microscope/home/index.php (accessed on 5 May 2022)) that allows for comparison with the NCBI databases. Strains 09-003 and 09-027 were both closely related to *Halomonas alkaliphila*. Strain 19-001 was separated from the other two *Halomonas* strains and formed a monophyletic group with *Halovibrio variabilis*, a homotypic synonym of *Halomonas variabilis*.

Next, we assessed whether the four selected B_12_-producing strains encoded any B_12_-related enzymes. Subsequent genomic analyses identified the complete aerobic vitamin B_12_ biosynthetic pathway in all strains, corresponding to 17 enzymatic reactions from substrate uroporphyrinogen III to adenosylcobalamin (Appendix A). Most importantly, all the strains encoded the *cobNST* gene complex considered a critical step for B_12_ synthesis [14]. To better characterize B_12_ dependency patterns of the selected bacteria, cobalamin-dependent enzymes were searched for in the bacterial genomes. Prokaryotes can encode a large diversity of B_12_-dependent enzymes [56,57,58,59], but the principal factor for B_12_ auxotrophy is the presence of the *metH* gene and the absence of the *metE* isoform [60]. A B_12_-dependent methionine synthase (*metH*; E.C. 2.1.1.13) was identified in all four strains. Unlike *P. porticola*, the three *Halomonas* spp. strains encoded an additional B_12_-independent methionine synthase (*metE*; E.C. 2.1.1.14). Moreover, all strains of both species encoded several cobalamin-dependent ribonucleotide reductase gene copies (E.C. 1.17.4.1) and no vitamin-independent isoforms, together with a *BtuC* permease allowing the uptake of environmental B_12_ into the cell [61]. These results indicate that the *Halomonas* strains have the same B_12_-related genetic portfolio. For the following experiments, we selected the bacterial strains that had the highest vitamin B_12_ content, namely *P. porticola* 09-029 and *Halomonas* sp. 19-001. In addition, we aimed to understand whether *T. lutea* in co-culture with bacteria from different families would lead to different interactions.

### 3.3. T. lutea–Bacteria Co-Cultures

We tested whether *T. lutea* would be able to access cobalamin by cultivating it either with bacterial extracts or in co-culture with live bacteria, in addition to B_12_-deprived and B_12_-enriched axenic control cultures. To test the ability of the alga to scavenge cobalamin from bacterial extracts, carbon-normalized extracts of the B_12_-producing strain *P. porticola* and of *Halomonas* sp. 09-729 as negative control (see Figure 1) were added in subcultures from the cobalamin-deprived axenic controls sampled at day 35 (Figure 2A). Measured vitamin concentrations in the bacterial extracts were 69.96 ± 5.23 ng/L B_12_ for *P. porticola* and 4.01 ± 1.41 ng/L B_12_ for *Halomonas* sp. 09-729, both in equivalent final concentration in *T. lutea* cultures. Interestingly, adding *P. porticola* bacterial extract to B_12_-deprived medium allowed *T. lutea* to develop to the positive control level (Figure 2A), demonstrating that *T. lutea* was able to scavenge cobalamin from the bacterial extract. In contrast, cultures with extract from the negative control strain 09-729 exhibited a slight growth that can be explained by a very low cobalamin content. According to our screening results (Figure 1), we assume this cobalamin originates from the assimilation of vitamin from the bacterial culture medium and not from a *bona fide* B_12_ biosynthetic ability.

In parallel, *T. lutea* was grown in B_12_-deprived and B_12_-enriched mixed cultures either with *P. porticola* 09-029 or *Halomonas* sp. 19-001 (Figure 2A). In addition to axenic controls, xenic cultures with cobalamin input were established to identify if bacteria could have a negative effect on *T. lutea* growth. We found that microalgal growth in co-culture with B_12_ was comparable to axenic controls (Figure 2A). 

The addition of *Halomonas* sp. 19-001 or *P. porticola* 09-729 in the presence of the vitamin did not lead to differential maximal biomass increase (ΔC_max_, Table 1, Appendix A) compared with axenic + B_12_ controls.

Grown in mixed B_12_-deprived cultures, none of the bacteria tested induced *T. lutea* growth compared with the axenic controls, which was illustrated by an absence of significantly different ΔC_max_ (*p*-value > 0.05) (Figure 2A, Table 2, Appendix A). As these bacteria were formerly selected for their high B_12_ content, this result indicates that they did not release enough vitamin to support *T. lutea*. Co-cultures were maintained for almost 50 days without growth increase, suggesting that bacterial lysis did not occur in the late stationary phase, which would have theoretically allowed the microalga to obtain cobalamin. For both bacterial strains, the bacteria/alga ratio in the stationary phase was low in the presence of vitamin, while it reached higher values in its absence (Figure 2B).

We observed that bispecific co-cultures of B_12_-producing bacteria were unable to lift the cobalamin deprivation, while the microalga was able to scavenge cobalamin originating from bacterial lysis. In a final experiment, we investigated whether a complex natural microbial community would promote the growth of *T. lutea* cultivated under cobalamin deprivation. To this end, we considered xenic *T. lutea* cultures that have not been formerly purified and conducted an experiment with and without vitamin B_12_ addition. Compared with B_12_-deprived axenic controls, cultures with the microbial community displayed higher growth, with six times superior ΔC_max_ (two-tailed Student’s *t* test; *p*-value = 7 × 10^−4^) (Figure 3, Table 3). Compared to xenic treatments without B_12_, the axenic controls with 40 ng/L cobalamin showed ~5.5 times superior maximal biomass increase, indicating that the microbial consortium synthesized ~7 ng/L cobalamin. Xenic treatments with cobalamin addition showed growth parameters equivalent to those of the positive controls, indicating an absence of negative interaction in more B_12_-replete conditions.

The microbial consortium composition of the xenic-B_12_ cultures was then assessed by metabarcoding (Figure 4). After bioinformatic processing, a total of 198,349 16S rDNA gene sequences were generated, of which 389 were present in only one replicate and not considered for further analysis. 

The remaining 197,960 sequences (61,892; 67,203; and 68,865 sequences for replicate 1, 2, and 3, respectively) were clustered into a total of 46 ASVs (41, 40, and 44 ASVs for replicate 1, 2, and 3, respectively) (Appendix A). Two ASVs remained unassigned at the phylum level. The global composition of the self-selected natural microbiome associated with *T. lutea* was characterized by the dominance of Firmicutes (Bacillales ~50% of the total reads, 4 ASVs) from the genus *Bacillus* and Spirochaetota (Spirochaetales, ~30%; 2 ASVs) from the genus *Alkalispirochaeta*. Proteobacteria also contributed significantly to the bacterial community (~15%; 26 ASVs). These consisted of Alphaproteobacteria (~12%; 17 ASVs), of which mostly Sphingomonadales (~10%; 4 ASVs) mainly from the genus *Porphyrobacter* (~8%; 1 ASV), with minor contributions of Rhodobacterales and Caulobacterales (~1% each; 5 and 4 ASVs, respectively). Other prokaryotic taxa such as Actinobacteria and Bacteroidetes contributed to ~1–2% of the total reads.

### 3.4. Functional Prediction of De Novo Cobalamin Synthesis in the Microbial Consoritum

We then investigated the potential metabolic pathways associated with the microbial consortium using the PICRUSt2 software, focusing the analysis on the pathway related to cobalamin synthesis from uroporphyrinogen III. A total of 31 ASVs displayed at least one of the reactions involved in this biosynthetic process (Appendix A), and we found that the microbial consortium gathered 15 of the 17 genes necessary to synthesize vitamin B_12_ de novo (Appendix A), equivalent to 14 enzymatic reactions (EC numbers). The remaining two genes (*cobA* and *cobL*) encoding the enzymes responsible for the reactions EC 2.1.1.107, 2.1.1.132 and 2.5.1.17 were absent from the MetaCyc reference database. One Actinobacterium (genus *Rhodococcus*; ~0.8% total reads) and two Rhodobacterales (genera *Sulfitobacter* and *Pseudophaeobacter*; ~0.1 and 0.01%) were predicted to encode the 14 enzymatic reactions. Conversely, the two most abundant ASVs corresponding to the *Bacillus* and *Alkalispirochaeta* genera did not encode 3 and 12 of those reactions, respectively (Appendix A).

## 4. Discussion

Limitation of phytoplankton development by vitamin B_12_ has been observed in different coastal and oceanic systems [6,8,9,62]. Only select species of bacteria and archaea are able to perform the vitamin B_12_ synthesis, which requires a complex set of around 20 genes [2,11]. Following a recent surge of interest [63], interactions between microalgae and bacteria for B_12_ have been unevenly investigated for important microalgal groups such as chlorophytes [15] and dinoflagellates [17,21]. Haptophyte microalgae, which are suspected to constitute an exclusively B_12_ auxotrophic taxon [22], are also ubiquitous and important contributors to primary productivity [23,28,64] and present a diverse range of interactions with bacterial communities [65,66]. Moreover, the interactions between B_12_-producing microbes and B_12_-dependent haptophytes have been suspected to be involved in the global sulfur cycle through DMSP production [19]. However, information about their B_12_ acquisition mechanisms remains elusive. Considering this gap in the knowledge, we attempted in the present work to characterize the way by which haptophytes may acquire vitamin B_12_ from bacteria, by focusing on the model species *Tisochrysis lutea*.

Bacteria were screened to identify strains with the ability to produce high vitamin B_12_ amounts. Two strains of either *Halomonas* sp. or *Phaeobacter porticola*, which showed the highest vitamin content, were grown with *T. lutea* with different B_12_ inputs to test whether a mutualistic relationship would occur in B_12_ deprivation. In [4], the authors experimentally demonstrated that the strain *Halomonas* sp. 19-001 we used here could sustain the dinoflagellate *Amphidinium operculatum* and the rhodophyte *Porphyridium purpureum* with enough B_12_, indicating that it has the ability to export cobalamin in a mutualistic relationship. Here, B_12_ production of this bacterium was confirmed but it appeared that this strain was unable to deliver enough cobalamin to *T. lutea* in our culture conditions. The second tested bacterium was *P. porticola* from the *Rhodobacteraceae* family of which members have been shown to directly provide B_12_ to the chlorophyte *Ostreococcus tauri* and the dinoflagellate *Lingulodinium polyedrum* [17,21]. Both experimental and in silico analyses provided evidence that *P. porticola* had the ability to synthesize the vitamin. In the same way as with *Halomonas* sp. 19-001, the experimental co-cultures with *P. porticola* showed that this bacterium did not allow *T. lutea* to mitigate B_12_ deprivation. The growth of *T. lutea* was similar between xenic and axenic cultures, demonstrating that none of the bacterial strains tested had a negative effect on the microalga. Contrary to the existing evidence that bacteria can directly deliver enough cobalamin to some microalgal taxa, we did not observe differential growth of *T. lutea* when grown with two B_12_-producing bacterial strains, suggesting that the conditions required for mutualism were not fulfilled. Moreover, the autotrophic production of B_12_ seems to be more crucial for *Phaeobacter*, which does not encode *metE* and hence is a B_12_ prototroph, than for *Halomonas*. Consequently, while the *Halomonas* strain could potentially grow without the vitamin, *P. porticola* must rely on its own vitamin production or that of other microbial community members to develop, which could potentially reduce the cobalamin quantity available for *T. lutea* in co-culture.

One major issue is how many bacteria are needed to sustain microalgal growth. Several studies [15,17,21] observed bacteria/algae ratios ranging from 1:1 to 400:1 for mutualistic algal-bacterial co-cultures for cobalamin. This discrepancy can be explained by the large difference in size between the studied species. A much narrower range is obtained when considering algal carbon content, that is from 0.1 to 1.5 bacterial cell/pg C (see Appendix A) [16,67,68,69,70,71]. A rough estimate of how many bacteria microalgae need can be obtained from the B_12_ contents of both microorganisms. For *T. lutea*, given that 40 ng/L of cobalamin leads to 25 × 10^6^ cells/mL (Figure 3), its B_12_ minimum quota is about 1.6 ag/cell. This corresponds to 67 µL of algal biomass per pmol of vitamin, in line with the yield requirement given in [72], i.e., from 30 to 100 µL/pmol, based on observations for several algal species. Now considering the bacterial B_12_ content that we measured (Figure 1), we obtained a theoretical bacteria–algae ratio of 10:1, or equivalently 1 bacterial cell/pg C. This estimation is similar to what has been observed experimentally with other species in the aforementioned studies. In the bispecific co-culture experiments, the bacteria/algae ratios were about 10:1 for *Halomonas* sp. 19-001 and hence supposedly sufficient to allow mutualistic growth of the microorganisms. For *P. porticola*, the ratio was lower, but still within the range of ratios observed in the literature, when expressed in C content. Aside from the yield, we assessed that the vitamin B_12_ amount theoretically made available by these bacteria with the measured biomass was ~2 to 4 ng/L B_12_ in co-culture with either *Halomonas* sp. or *P. porticola*, respectively, assuming that all microbial B_12_ would be excreted in the medium. These results correspond to very low cobalamin concentrations, equivalent to the ones measured in the Southern Ocean [73], the Indian Ocean [74], and the Sargasso Sea [75] where auxotrophic phytoplankton is cobalamin-limited. All of these estimates are based on the bacterial production of B_12_ measured in the screening experiment (Figure 1). The literature provides conflicting results as to whether or not this content depends on environmental conditions. A previous study reported that *Halomonas* spp. isolated from natural communities did not encode a B_12_ riboswitch, resulting in an inability to regulate their vitamin production [13], hence potentially producing elevated B_12_ in the extracellular medium. This observation makes the absence of significant growth of *T. lutea* in co-culture with *Halomonas* unclear. On the other hand, all of the bacteria investigated can acquire B_12_ from their surrounding environment, suggesting an ability to repress the costly cofactor synthesis [76]. Aside from this uncertainty about the regulation of vitamin production, one explanation could be that B_12_, if produced, was not excreted. 

The absence of positive bacterial effects raises the question of another parameter controlling algae–bacteria interactions. For instance, the production of infochemicals has been evidenced as a way for microorganisms to interact in the phycosphere, the specific microenvironment surrounding algal cells. Such chemical communication has been reported several times for the haptophyte *Emiliania huxleyi* and its associated bacteria. As an example, [77,78] described an interaction model in which senescent *E. huxleyi* produce p-coumaric acid, stimulating the excretion of roseobacticides, molecules that are detrimental to *E. huxleyi*, by *Phaeobacter gallaciensis* (*Roseobacteraceae*). Another study observed that the production of bacterial indole-3-acetic acid mediates the interaction between *Phaeobacter inhibens* and *E. huxleyi* [65]. *P. inhibens* was in turn able to assimilate DMSP produced by the microalgae as a sulfur source and incorporate DMSP degradation products into amino-acids and roseobacticides. Most studies analyzing interactions between *E. huxleyi* and species from the *Phaeobacter* genus reported a switch from mutualism to a pathogenic relationship. Here, co-cultures of B_12_-producing bacteria and the haptophyte *T. lutea* did not reveal mechanisms of interactions leading to bacterial lysis and/or the release of bacterial B_12_ allowing the algal growth, nor to microalgal cell death even after 50 days in culture. It seems plausible that signal molecules came into play in previously observed B_12_ co-cultures, and particularly the ones involving the *Halomonas* 19-001 strain we used here [4]. This could indicate interaction specificity and explain why adding this bacterium did not allow *T. lutea* to access cobalamin. 

In contrast with bispecific co-culture experiments, cultivating *T. lutea* with a natural microbial consortium (i.e., prokaryotes that were selected from the original microbial population and that were maintained in the laboratory conditions since the strain isolation) partially compensated for the absence of cobalamin. Although the microbial community was composed of some taxa that are known to produce vitamin B_12_, such as the *Bacillus* genus of which species *B. megaterium* is used for industrial production of the vitamin [2], and Rhodobacterales [14], we are unable to conclude on whether it is one taxon or a combination of several taxa that made cobalamin available to *T. lutea*. Division of labor among members of the same microbial community has been proposed as a way to maximize benefits at the community level. A thought-provoking example is the study of [13] which demonstrated how a complex community composed of many auxotrophic microbes can be self-sustained in a medium lacking essential cofactor, including cobalamin, by combining a mosaic of precursor salvage pathways and transporters. Such a collaboration seems conceivable in our experiment, especially when considering the results of the functional predictions showing that 67% of the ASVs we identified likely encoded at least one enzymatic step of B_12_ biosynthesis. In addition, some microorganisms, including microalga, are able to remodel pseudocobalamin, a significantly less available cobalamin isoform produced by cyanobacteria, into a metabolizable vitamin [79]. It is therefore plausible that members of the microbial community could have salvaged and metabolized precursors, leading at some point to cobalamin excretion. Further and thorough characterization of the community members would be required to conclude on their essential metabolic requirements and biosynthetic potential.

On the other hand, we observed that *T. lutea* was able to scavenge and assimilate cobalamin-rich bacterial lysate, compensating B_12_ limitation, indicating that bacterial lysis in the environment may provide cobalamin to some microalgae. A recent study provided evidence that viral lysis of heterotrophic bacteria can release nutrients that are then taken up by marine phytoplankton [80]. This may constitute a way for B_12_ auxotrophs to obtain the vitamin and implies a potential ecological succession of haptophytes and B_12_-producing bacteria rather than a direct positive interaction of the organisms. Moreover, we previously observed that dissolved methionine may be assimilated by *T. lutea* when grown in cobalamin scarcity, allowing the microalga to develop, but not to the control + B_12_ levels [22]. While we cannot rule out the possibility that *T. lutea* grown with bacterial extracts would have taken bacterial methionine up, this would have been insufficient to account for the important growth we observed. Consequently, the availability of bacterial B_12_ in the extract is the parameter that best explains the growth recovery. 

A previous study [81] reported selective feeding of the mixotrophic chlorophyte *Micromonas* spp. and of the haptophyte *Isochrysis galbana*, a species closely related to *T. lutea* [40], in B_12_-deprived cultures, allowing the algae to survive by increasing bacterivory. As mixotrophy is an important characteristic of haptophytes [25,81,82,83,84], we suggest that phagotrophy of bacteria, or bacterivory, may constitute an alternate way to obtain cobalamin at least for some haptophyte species. It might be that the species of bacteria tested here were not edible by *T. lutea* or that the microalgae did not encode the adequate genetic portfolio to perform phagotrophy. The hypothesis that haptophyte may indirectly assimilate B_12_ by bacterivory should be tested with representatives from different haptophyte families and at different vitamin availability levels. This could also be worth investigating for dinoflagellates, the other B_12_-dependent group that is known to include phagotrophs.

## 5. Conclusions

Previous studies have demonstrated that for some microbial associations, a mutualistic relationship for cobalamin may occur, in which microalgae trade organic substrates in exchange for bacterial B_12_ [15,16,17,21]. The present work is, to our knowledge, the first testing B_12_ acquisition by haptophytes and shows that not all bacteria share their vitamins directly, or at least not with all algae. A lift of B_12_ deprivation was observed only when the algae were grown with a complex microbial community predicted to encode the vitamin B_12_ biosynthetic pathway, paving the way for new research avenues involving metabolic complementation between microbes for the benefit of both eukaryotic and prokaryotic members. Moreover, our results illustrate that to some extent *T. lutea* is able to scavenge the vitamin from bacterial lysate, a strategy that could involve an ecological succession of haptophytes and B_12_-producing bacteria in environmental plankton communities. While this study yielded important insights regarding interactions between haptophytes and bacteria for cobalamin, further analyses are needed on a diverse range of haptophyte species to better characterize biotic interactions of this important microalgal community with bacteria and archaea.

## Figures and Tables

**Figure 1 microorganisms-10-01337-f001:**
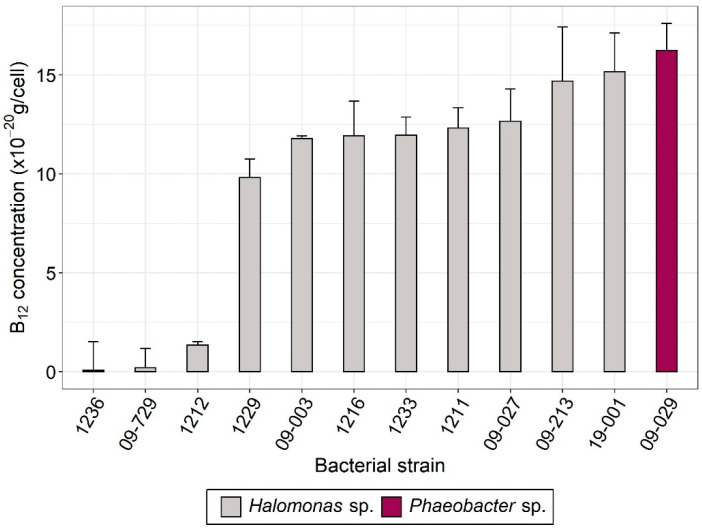
Screening of 12 bacterial strains for vitamin B_12_ content (means of *n* = 9 technical replicates ± standard deviation of the mean). The genus of the isolates is indicated.

**Figure 2 microorganisms-10-01337-f002:**
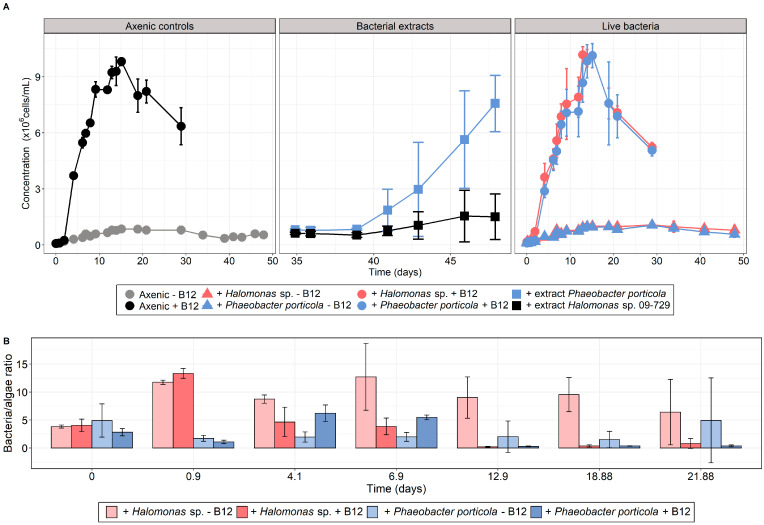
Cultivation experiments of *T. lutea* with different B_12_ sources. (**A**) Growth curves for axenic controls (left panel), with bacterial extracts (center panel), or in co-culture with single live bacterial strains (right panel), with different B_12_ availability levels (means of *n* = 3 biological replicates ± standard deviation of the mean). (**B**) Bacteria/algae ratios for the co-cultures with live bacterial cells with different B_12_ availability levels, as above (means of *n* = 3 biological replicates ± standard deviation of the mean).

**Figure 3 microorganisms-10-01337-f003:**
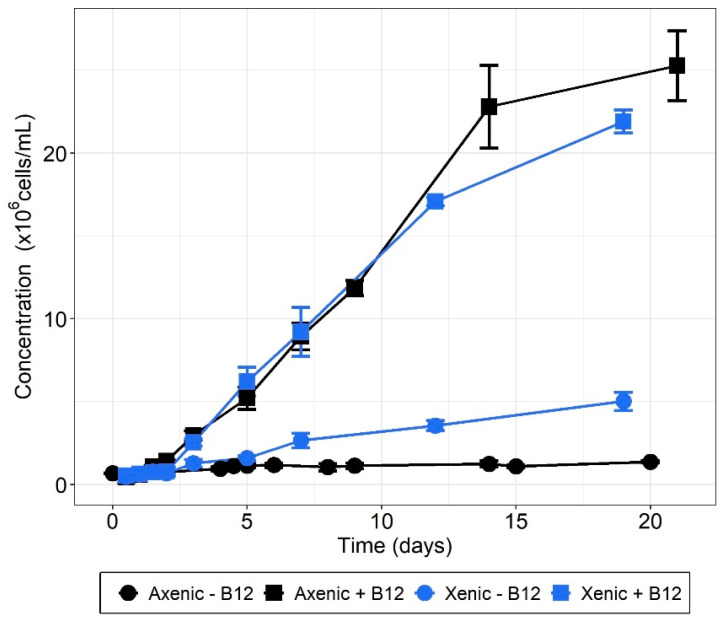
Co-culture experiment of *T. lutea* with a complex microbial consortium with different B_12_ availability levels (means of *n* = 3 biological replicates ± standard deviation of the mean).

**Figure 4 microorganisms-10-01337-f004:**
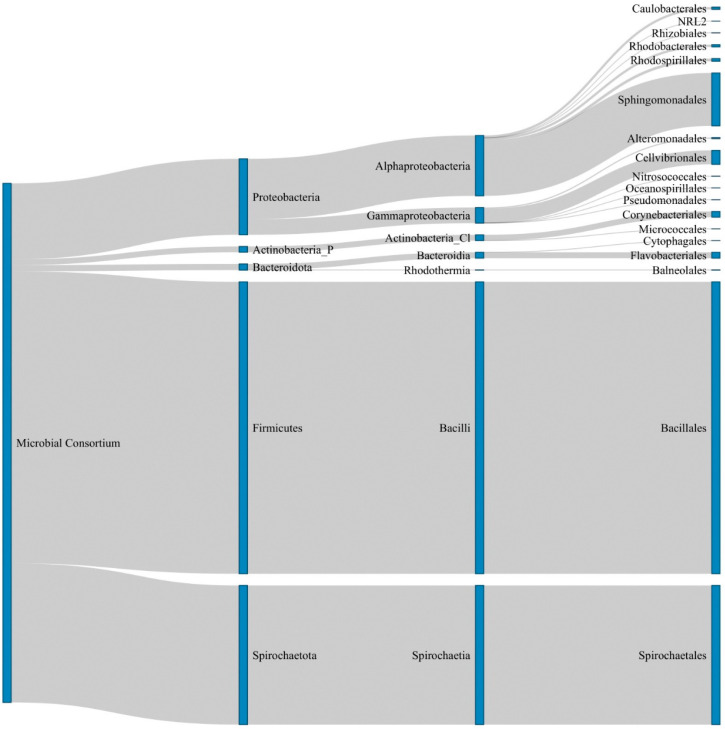
Sankey diagram representing the composition of *T. lutea*-associated microbial consortium (xenic-B_12_) estimated by metabarcoding. From left to right are the relative proportions of taxa at the phylum, class and order levels for one of three replicates (see Appendix A for details on the other replicates) using the ASVs assigned starting at the phylum level. Actinobacteria_P: phylum Actinobacteria; Actinobacteria_Cl: class Actinobacteria.

**Table 1 microorganisms-10-01337-t001:** Origin of the bacterial strains screened.

Strain Name	Species	Origin
09-029	*Phaeobacter* sp.	*Odontella aurita* and *Emiliania huxleyi* [31]
09-003	*Halomonas* spp.	
09-027		
09-213		
09-729		
1211	*Halomonas* spp.	Deep-sea ecosystems
1212		
1229		
1233		
1236		
19-001	*Halomonas* sp.	*Amphidinium operculatum* [4]

**Table 2 microorganisms-10-01337-t002:** Maximal biomass increase of *T. lutea* culture experiments with live bacteria or bacterial extracts, with or without cobalamin input.

Culture Condition	ΔC_max_ (10^6^ Cells/mL)
Axenic − B12	0.86 ± 0.10
Axenic + B12	9.73 ± 0.22
19-001 − B12	0.97 ± 0.06
19-001 + B12	10.23 ± 0.21
09-029 − B12	0.98 ± 0.12
09-029 + B12	10.03 ± 0.65
09-029 extract	6.81 ± 1.59
09-729 extract	0.96 ± 1.35

**Table 3 microorganisms-10-01337-t003:** Maximal biomass increase of *T. lutea* culture experiments with a complex microbial consortium, with or without cobalamin input.

Culture Condition	ΔC_max_ (10^6^ Cells/mL)
Axenic − B12	0.69 ± 0.07
Axenic + B12	24.86 ± 2.16
Xenic − B12	4.54 ± 0.72
Xenic + B12	21.37 ± 0.75

## Data Availability

The four bacterial genomes sequenced in the present study are openly available under the accession identifiers: “*Halomonas alkaliphila* PBA_09_003”; “*Halomonas alkaliphila* PBA_09_027”; “*Halomonas* sp. PBA_19_001”; and “*Phaeobacter porticola* PBA_09_029” at the MicroScope platform (https://mage.genoscope.cns.fr/microscope/home/index.php (accessed on 5 May 2022)).

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
