# Peer review of "Sharing Vitamin B12 between Bacteria and Microalgae Does Not Systematically Occur: Case Study of the Haptophyte Tisochrysis lutea"

_microorganisms, 2022, doi:10.3390/microorganisms10071337_

Round 1
Reviewer 1 Report
The authors investigated bacteria algae relationships based on the process of vitamin B12 acquisition providing also genomic data. This topic is highly interesting also in an industrial point of view since we know that the microbiome can influence algal growth performance. The article is well written, and the results are adequately compared and discussed with the literature.
Specific comments:
Title is too long, please be more concise
Lines 126 and 333: rRNA should be changed to rDNA, please check the manuscript and supplemental material
Line 138: please format the units consistently, e.g., µmol photons m-2 s-1
Line 145: please format citation Walne, 1970 and add to reference list according to the journal standards
Lines 77, 139 and 230: change "microalgae cultures" to "microalgal cultures"
Line 170: What does the Sybr Green stain? Reference?
Line 248: How the species 09-029 was identified? The species level is not clear from the tree shown in the supplemental Figure S1. Supplemental Figure S1 needs to be improved, the tree is very difficult to read in the current form.
Line 385: microalga needs be in singular or substituted by T. lutea
Line 497: Here, you are discussing the microscopic results but they are not presented in the article. Please add microscopic pictures to the article or at least to the supplemental data to show the xenic and axenic cultures.
English corrections: line 22 "show" should be in the past "showed", line 492 delete “the”, line 504 microalga in singular
Reviewer 2 Report
Review comments on “Sharing vitamins, or not? Co-cultures of two known vitamin B12-producing bacteria does not allow growth of B12-starved Tisochrysis lutea, but its natural microbial consortium does”
General comment:
3.3. T. lutea–bacteria co-cultures: This experiment was so carefully designed and executed, which is really very crucial and commendable. 35 days starved culture and then progressing for another 30 days with and without B12 sources is really nice.
Specific comments:
Lines122-124: Provide the electronic link/source for these genomes in parenthesis.
Line126: 2.3. Phylogenetic analysis of the bacteria: Provide the sequence size that were used to construct the phylogenetic trees.
Line144: Provide the appropriate reference for the axenic culture preparation.
Supplementary Table S3: It would be nice if column 1 can be moved to Column 6 for the references.
Reviewer 3 Report
The manuscript comes with a study of the acquisition process of B12 vitamin by haptophyte (Tisochrysis lutea). The authors tested the effects of bacterial extracts, co-culture (bacteria/algae) and complex bacterial consortium on the growth of microalgae.
Through the obtained results, authors concluded about the indirect acquisition of B12 vitamin by microalgae in complex natural bacterial communities since the growth of tested microalgae was partially rescued using bacterial consortium (compared to co-culture bacteria/algae).
In all, the manuscript is interesting; however, it has to be improved in order to be suitable for publication.
Recommendations
To improve the manuscript, authors should consider the following recommendations:
- The title of the manuscript should be reconsidered by giving indication of the main obtained results through the conducted study.
- Authors cited in “Material and methods” section several bacterial strains from different origins whereas only strains from marine microalgae culture, Phaeobacter and Halomonas sp. 19-001 were used in the study??
Since the strains from deep sea ecosystems were not tested during this work (tested only for vitamin B12 content), their citation could be avoided.
- Besides, Line 97 “Four bacterial strains were selected (3 Halomonas / 1 Phaeobacter) for genome sequencing”. A justification of the choice of those bacterial strains must be provided.
- Line 88 “Particular vitamin B12 extraction from bacterial culture was conducted”. A brief description of the main extraction steps should be cited.
- Authors added vitamin B12 in the form of cyanocobalamin and choose 40 ng/L as concentration. Based on what the choice was made??
- Tisochrysis lutea was used in the conducted study. Author should indicate the origin of the microalgae like for the bacterial strains.
